# Simulation Training Needs of Nurses for Nursing High-Risk Premature Infants: A Cross-Sectional Study

**DOI:** 10.3390/healthcare10112197

**Published:** 2022-11-02

**Authors:** Sun-Yi Yang

**Affiliations:** College of Nursing, Konyang University, Daejeon Medical Campus, 158, Gwanjeodong-ro, Seo-gu, Daejeon 35365, Korea; lavender799@gmail.com; Tel.: +82-42-600-8560

**Keywords:** nurses, infant, premature baby, intensive care, neonatal, clinical practice

## Abstract

Opportunities fomr clinical training in the newborn nursery and neonatal intensive care units (NICU) are becoming insufficient and limited to observation-oriented training. Premature infants admitted to the NICU require specialized and highly sophisticated care. Therefore, this descriptive cross-sectional study aimed to understand nurses’ educational needs for establishing a high-risk premature infant nursing simulation training program. It used a descriptive cross-sectional design. We conducted a survey involving 99 newborn nursery and NICU nurses; data were analyzed using descriptive statistics, a paired *t*-test, an Importance-Performance Analysis (IPA), a Borich needs analysis, and the locus for focus to confirm educational priorities. The scores indicating the importance of nursing high-risk premature infants were higher than those of performance. Results indicated that the highest educational need was in the “treatment and procedure” domain. After deriving the priority of educational needs using the Borich needs analysis and the locus for focus, “maintenance of various tubes” showed the highest priority. By identifying the training priorities for high-risk premature infants nursing using various analytical frameworks, an extended reality simulation program met nurses’ high-risk premature infant nursing educational needs. Training for high-risk premature infants treatment and procedures—maintaining various tubes—is highly demanded by nurses and should be prioritized.

## 1. Introduction

Premature infants are defined as those born before the 37th week of pregnancy and their number has been increasing due to reasons such as the aging of mothers, increased high-risk pregnancy and assisted reproduction, and advances in medical technology and equipment [1,2]. Globally, premature infants account for 11% of all births, at 15 million each year; of these, 10–15% are treated and nursed in neonatal intensive care units (NICU) [3,4]. The survival rates of premature infants undergoing treatment and nursing in NICUs are gradually increasing but the long- and short-term morbidity and mortality rates remain high [1,4].

Nursing premature infants is a highly professional and complicated task. As their prognosis can differ significantly, depending on the passage of treatment time, the status of premature infants must be quickly and accurately assessed and specialized nursing interventions must be provided [2,3]. In particular, respiratory issues commonly occur in premature infants due to insufficient pulmonary surfactants, and healthcare workers’ ability to respond to emergencies is directly related to their survival and prognosis [5]. Accordingly, there is an increasing demand for professional knowledge and skills of NICU nurses [6].

However, the global outbreak of the coronavirus disease (COVID-19) pandemic limited or suspended NICU clinical training for high-risk neonatal infection control. Moreover, even in cases where clinical training is allowed, the opportunities to perform direct nursing has gradually decreased and the training has been limited because of concerns regarding infant patient safety [6,7,8,9]. NICU requires professional knowledge and skills regarding special equipment and nursing premature infants; however, owing to a lack of nurses with abundant experience and competence, high-risk neonatal nursing is not being provided adequately and NICU nurse training is insufficient [10,11].

As measures to overcome these limitations, the development of various non-face-to-face simulation training programs covering virtual, augmented, and mixed reality is being reinvigorated [6,12,13,14]. Simulation training is a significant educational method that connects knowledge, skills, and attitudes while ensuring patient safety in a virtual environment [15,16]. Another study [4] also notes that the utilization of new technology is anticipated to play an important role in promoting nursing education and reducing training time. Thus, it can serve as an alternative or complementary method to traditional clinical training.

However, a recent systematic literature review [12] shows that most of the currently available NICU simulation programs focus on education for nursing college students, medical students and interns, and residents, with the educational content limited to a neonatal resuscitation program (NRP) and respiratory distress syndrome. Swift nursing treatment must be provided for high-risk premature infants at birth, for which nurses must have the proficiency to respond quickly and accurately to emergencies, based on abundant experience and intuitive thinking [17]. Previous studies [17,18,19] focus mainly on identifying the necessity of nursing work in special situations, or the necessity of education in certain nursing fields. Nurses in the NICU feel that more specified education is needed for continuous development of expertise; however, they note that it is difficult to find studies that grasp the training contents required by nurses, an all-encompassing clinical nursing training [20].

In “Importance-Performance Analysis,” (IPA) Martilla and James [21] note that a process is needed to identify the difference between the current level and status of education and the expected level and status thereof, that must be addressed through education. The priorities in educational needs can be understood by identifying IPA among nurses at a clinical site. Furthermore, the differences in nursing performance can be analyzed and diagrammed into quadrants using Borich’s [22] needs assessment model and the locus for focus model by Mink et al. [23]. This process of analysis has also been verified by various previous studies [24,25,26] to play the main role in identifying the educational requirements of nurses. As such, this study sought to provide basic material for the development of NICU extended reality (XR)-based simulation programs by identifying the training status of nurses in newborn nursery and NICU, to facilitate nursing of high-risk premature infants. It investigated the educational requirements and their importance based on the NICU nursing classification system by utilizing various analytical frameworks (IPA, Borich needs assessment model, the locus for focus model) to identify nursing training priorities for high-risk premature infants.

This study aimed to identify these priorities for the following specific purposes. First, the newborn nursery and NICU nurses’ perception of the difference in the importance and performance of nursing high-risk premature infants was identified. Second, the variance in the perceived importance and performance of nursing high-risk premature infants was determined using the IPA. Third, the training needs and priorities of the nurses were identified utilizing the needs assessment model by Borich [22] and the locus for focus model by Mink et al. [23].

## 2. Materials and Methods

### 2.1. Research Design

This study used a descriptive cross-sectional design to understand the training needs of newborn nursery and NICU nurses for the development of XR-based nurse training programs for nursing high-risk premature infants in the NICU.

### 2.2. Research Participants

This study sample comprised newborn nursery and NICU nurses. A recruitment notice was sent to hospitals with NICU and requested to be posted on the NICU bulletin board. Those who wished to participate in the study could access the questionnaire by scanning a QR code. The selection criteria included nurses who had worked or were working as nurses in a newborn nursery or NICU for more than one year. Both male and female nurses were included. In terms of the exclusion criteria, nurses who had no experience of working in a nursery or NICU were excluded as their inclusion could affect the study results. Nurses who had previously participated in a NICU Nursing Training XR program were also excluded. The survey was conducted online from May to June 2022.

The number of study participants was calculated using G*power 3.1.9 [27], a sample size calculation program. At small effect size, (d) = 0.3, was used in the paired *t*-test to analyze differences in the importance and performance of nursing for high-risk premature infants, significance level (α) = 0.05, and the power (1 − β) = 0.80. The resultant minimum sample size was 90. The dropout rate was 0.7–11.6% in previous studies [2,26,28] and 1.0% in this study. Of the 100 respondents who were selected via convenience sampling, 1 provided an incomplete survey, making the final sample size 99.

### 2.3. Research Tool

#### 2.3.1. The Importance of Nursing High-Risk Premature Babies

This study used the Korean Patient Classification System for Neonatal Care Nurse (KPCSN) developed by Yu et al. [29] through reliability and validity verification with the support of the Korean Hospital Nurses Association, to measure the importance of nursing high-risk premature infants. The current published version of this scale quantifies nursing needs. We modified the original scale to capture how important nurses found the various practices. As an example, the question “Infant needs to be prepared for tracheostomy insertion” was changed to “How important is it to prepare the infant for tracheostomy insertion?” To accomplish this, we replaced the term “need” with “important” in the questions and answer options of each question. Therefore, questions were rated on a Likert-type scale from “very important” to “not important at all.” Prior to using the scale for the present study, five experts (two NICU nurses and three professors of child health nursing) validated these modifications where I-CVI > 0.8.

This tool comprised 71 questions, including those on monitoring and measuring of vital signs (9 questions); physical examination and test (8 questions); respiratory care (13 questions); mobility (2 questions); hygiene and infection control (5 questions); feeding (2 questions); elimination (3 questions); medication and transfusion (7 questions); treatment and procedure (15 questions); emotional support, communication, and education (2 questions); and 11 sub-components of admission and discharge management (5 questions). In each question, the importance was measured on a five-point Likert scale ranging from “very low” (one point) to “very high” (five points). The scale scores were calculated as the item score average and presented as an average value ranging from 1–4, where the higher the score, the more important the nursing of high-risk premature infants was recognized. At the time of tool development, the verification result of conformity between nurses and nursing managers for reliability verification was *r* = 0.58–0.87 [29], with the Cronbach’s *α* in this study at 0.94.

#### 2.3.2. Performance in Nursing High-Risk Premature Babies

To assess the performance in nursing high-risk premature infants, this study modified the same KPCSN scale developed by Yu et al. [29] that originally quantifies nursing needs by replacing “need” with “performance” in each question. As an example, the question “Infant needs to be prepared for tracheostomy insertion” was changed to “How do you perform the infant’s preparation for tracheostomy insertion?” Prior to using it for this study, these modifications were validated by the same five experts (where I-CVI > 0.8) who validated the modification of the KPCSN scale to measure “importance” of the various nursing practices for high-risk premature infants.

The tool comprised the same 71 questions as those in the tool used to measure the “importance.” The performance in each question was measured on a five-point Likert scale, ranging from “very low” (one point) to “very high” (five points). The scale scores were calculated as the item score average and presented as an average value ranging from 1–4, where the higher the score, the higher the performance in nursing high-risk premature infants was considered. At the time of tool development, the verification result of conformity between nurses and nursing managers for reliability verification was *r* = 0.58 − 0.87 [29], with the Cronbach’s *α* in this study at 0.97.

#### 2.3.3. Importance-Performance Analysis (IPA) for Nursing High-Risk Premature Babies

The IPA developed by Martilla and James [21] was used to analyze the importance-performance of nursing high-risk premature infants. This method identifies items by dividing them into four quadrants, based on the *X*-axis (performance) and the *Y*-axis (importance). Quadrant I, labeled “keep up the good work,” is a desirable area indicating both high importance and performance of items. Quadrant II, the “concentrate here” area, comprises items with high importance but low performance that require immediate and the most intensive improvement. Quadrant III is the “lower priority” area with both low importance and performance. Quadrant IV is the “possible overkill” area with low importance but high performance items where a shift of resources must be considered.

#### 2.3.4. Analysis of Nursing Training Needs for High-Risk Premature Babies

In this analysis, Borich’s [22] priority formula in their needs assessment model is utilized to identify “performance,” which is the current level of competence (“what is”), and “importance,” which is the level of competence required (“what should be”), and the difference between the two is multiplied by the average importance as weight to determine priorities in educational needs. Borich’s priority formula is as follows:

Borich‘s needs ={∑(RCL − PCL) × Avg.(RCL)}/N (RCL = Required Competence Level, PCL = Present Competence Level, Avg.(RCL) = average score of importance by each competency, N = total number)

#### 2.3.5. Priority Analysis of Nursing Training Needs for High-Risk Premature Babies

Using the locus for focus model by Mink et al. [23], items were divided into four quadrants, based on the *X*-axis (required competency level average) and *Y*-axis (required competency level—present competency level average, degree of discrepancy between importance and performance), graphed, and identified. Quadrant I is an area with high discrepancy/high performance (HH), where the priorities greater than the average value for both importance and degree of discrepancy are the highest. Quadrant II is an area with high discretion/low performance (HL), where the importance is lower than the average value but the degree of discrepancy is higher and the priority to increase performance is second highest. Quadrant III is an area with low discretion/low performance (LL), where the priorities lower than the average value for both importance and degree of inconsistency are the lowest. Quadrant IV is a low discrepancy/high performance (LH) area, where the importance is higher and the degree of inconsistency is lower than the average value and the current performance is high.

### 2.4. Data Collection

For this study, research approval was obtained from K University Institutional Review Board after deliberation on the purpose, method, guarantee of the rights of the study participants, and questionnaire (IRB No. XXX).

For data collection, a recruitment notice and application link were e-mailed to the executive staff of the Hospital Neonatal Nursing Association website, and an official letter was sent to the hospitals operating NICUs. This study sought to ensure spontaneity by clearly stating in the participant recruitment announcement, agreement process, and online questionnaire that participation in the survey can be voluntarily suspended at any time without any consequent disadvantage. It was also explained that the survey data would only be used for research purposes and that anonymity and confidentiality were guaranteed. An online self-report questionnaire was distributed to those who expressed their intention to participate in the study; the questionnaire took approximately 30 min to complete.

### 2.5. Data Analysis

The collected data were analyzed using the statistical package PASW 27.0 for Windows (SPSS Inc., Chicago, IL, USA). The normality was tested using the Shapiro–Wilk test and the homoscedasticity was confirmed by the Levene test, upon which parametric analysis was used.

1. General characteristics and the importance of and performance in nursing high-risk premature infants were analyzed by frequency and percentage and mean and standard deviation. 

2. The differences in the importance of and performance in nursing high-risk premature infants according to general characteristics were analyzed by chi-squared test, Fisher’s exact test, independent *t*-test, and one-way analysis of variance ANOVA. 

3. The differences in the importance of and performance in nursing high-risk premature infants were analyzed by paired *t*-test and IPA to identify key areas with high necessity for improvement through education.

4. The Borich priority formula was measured using the Excel-based Mean Weighted Discrepancy Score Calculator to analyze nursing training needs for high-risk premature infants, and the degree of discrepancy between the importance of and performance in nursing high-risk premature infants was analyzed using the locus for focus model. 

5. The differences in the importance of and performance in nursing high-risk premature infants was analyzed by paired *t*-test, the Borich priority formula, and the locus for focus model to select the priorities in nursing training needs for high-risk premature infants. The overlapping items were derived by aggregating the IPA.

## 3. Results

### 3.1. General Characteristics of Participants

The age group of most participants was 30–40 years old (68.7%) and the most common final academic degree was a bachelor’s degree (82.8%). Most participants had clinical experience of between five to seven years (52.5%) and experience in the newborn nursery and NICU of between three to five years (33.3%). The proportion of nurses working in the nursery and NICU was 45.5% and 54.5%, respectively. As for the number of patients assigned, the greatest proportion of nurses (62.6%) was assigned one to three patients or fewer, with 37.4% assigned four or more patients. Most of the nurses (89.9%) worked 8 to 10 h per day on average, with 3.0% working 10 h or more. For guardian visiting hours, an average of 30 to 60 min per day accounted for the highest proportion of nurses (57.6%). Of the participants, 90.9% had experience in clinical nursing training for high-risk premature infants, while 9.1% did not have such experience. Of the locations for clinical nursing training, 53.3% were on-site and 46.7% were at external institutions (universities, hosted by academic societies, organized by the Association of Nurses). The training method comprised lectures for 32.2%, simulation for 30.0%, case-based learning for 26.7%, and problem-based learning for 11.1%. In terms of locations where the participants desired to receive clinical nursing training for high-risk premature infants in the future, 66.7% preferred on-site training and 33.3% external institutions. Confidence in nursing high-risk premature infants was perceived to be high by 69.7%, medium by 26.3%, and insufficient by 4.0%, while the need for XR-based neonatal nursing training programs was demonstrated to be 97.0%, with most nurses desiring it (Table 1).

### 3.2. Differences in the Importance of and Performance in Neonatal Intensive Care According to General Characteristics

The analysis of the differences in the importance of neonatal intensive care according to the general characteristics of nursery and NICU nurses demonstrated that the importance was perceived to be significantly higher among nurses in the older age group with longer clinical, nursery, and NICU experience. There were significant differences in importance where the working unit was the NICU rather than the nursery, where the nurses had received clinical nursing training for high-risk premature infants by lectures, and where the need for XR-based neonatal nursing training programs was high. Analysis of differences in neonatal intensive care performance according to general characteristics demonstrated that the performance was significantly higher among nurses in the older age group with higher academic degrees and with more clinical, nursery, and NICU experience. In terms of working unit, the performance was significantly higher among nurses in the NICU than those in the nursery (Table 1).

### 3.3. Importance-Performance Analysis (IPA) for Nursery and NICU Nurses in Nursing High-Risk Premature Babies

The degree of the nurses’ perception of importance regarding nursing high-risk premature infants in the nursery and NICU was demonstrating significant differences (paired *t*-test = 3.31, *p* = 0.001). In terms of perceptions of importance, the “feeding” domain was found to be the most important, followed by “mobility”, “physical examination and test”, “emotional support, communication, and education”, “respiratory care”, “medication and transfusion”, “hygiene and infection control”, “admission and discharge management”, “monitoring and measuring”, “treatment and procedure”, and “elimination” (Table 2).

In terms of performance, the “feeding” domain was the highest, followed by “physical examination and test”, “mobility”, “hygiene and infection control”, “admission and discharge management”, “respiratory care”, “medication and transfusion”, “emotional support, communication, and education”, “monitoring and measuring”, “elimination”, and “treatment and procedure” (Table 2).

In the IPA analysis, items corresponding to the “concentrate here” area comprised the following eight items: “assessment of fall, pain, pressure sores, and sedation” (No. 13); “blood sugar test” (No. 14); “application of nebulizer” (No. 24); “endotracheal extubation” (No. 29); “exchanging linen” (No. 34); “enema (the preparation of enema with glycerin and physiological saline solution, and injection)” (No. 42); “maintenance of various tubes” (No. 54); and “family counseling (direct and indirect breastfeeding education, family interviews, etc.)” (No. 65) (Figure 1).

### 3.4. Borich Analysis of Nursing Training Needs for High-Risk Premature Infants in NICU Nurses

Upon identifying the training needs of the newborn nursery and NICU nurses for each domain of nursing high-risk premature infants using Borich needs analysis, the training needs for the “treatment and procedure” domain were found to be the highest, followed by the “monitoring and measuring” domain, “emotional support, communication, and education” domain, “medication and transfusion” domain, “respiratory care” domain, “mobility” domain, “elimination” domain, “physical examination and test” domain, “feeding” domain, and the “hygiene and infection control” and “admission and discharge management” domains.

In terms of training needs by item, scores were the highest for “peritoneal dialysis” (No. 61) in the “treatment and procedure” domain, followed by “preparing and nursing for tracheostomy insertion” (No. 57) and “Intra-aortic balloon pump (IABP)” (No. 6) in the “monitoring and measuring” domain, and the “NO therapy” (No. 21) item in the “respiratory care” domain (Table 2).

### 3.5. Locus for Focus Model Analysis on the Nursing of High-Risk Premature Infants by NICU Nurses

The analysis of nursing training needs for high-risk premature infants among nursery and NICU nurses using the locus for focus model showed that the HH (high discrepancy, high importance) quadrant included the following six items: “intake and output check” (No. 7) in the “monitoring and measuring” domain; “blood transfusion” (No. 46) in the “medication and transfusion” domain; and “simple dressing” (No. 51), “maintenance of various tubes” (No. 54), “therapeutic thermoregulation” (No. 62), and “NRP” (No. 64) in the “treatment and procedure” domain. The HL (high discretion, low importance) quadrant comprised the following ten items: “maintenance of IABP and extracorporeal membrane oxygenation (ECMO)” (No. 6) in the “monitoring and measuring” domain; “initiation and exchange nitrate oxide therapy” (No. 21) in the “respiratory care” domain; “cases in which nelaton catheterization is performed” (No. 41) in the “elimination” domain; “exchange transfusion” (No. 47) in the “medication and transfusion” domain; and “prepare puncture (pericardium, pleural cavity, lumbar puncture)” (No. 53), “complex dressing” (No. 53), “preparing for continuous renal replacement therapy (CRRT) insertion” (No. 59), “CRRT maintenance care” (No. 60), “perform peritoneal dialysis” (No. 61), and “care of the dying patient” (No. 71) in the “admission and discharge management” domain (Figure 2).

### 3.6. Deriving Priorities in the Nursing Training Needs for High-Risk Premature Infants

To select priorities in the nursing training needs for high-risk premature infants, overlapping items were identified by aggregating the IPA, Borich priority formula, and the locus for focus model. Items of topmost priority included “maintenance of various tubes” (No. 54), while items of secondary priority included “simple dressing” (No. 51), “intake & output check” (No. 7), “therapeutic thermoregulation” (No. 61), “blood transfusion” (No. 46), “droplet and air containment nursing” (No. 37), and “NRP” (No. 64) (Table 3).

## 4. Discussion

This study aimed to identify differences and priorities in the training needs and performance of newborn nursery and NICU nurses for nursing high-risk premature infants.

The importance of nursing high-risk premature infants, as regarded by the nursery and NICU nurses was 3.78 (0.67) out of 5 points, while the performance was 3.69 (0.57) out of 5 points. The comparison was limited, as no prior studies had measured importance using the same measurement tool; however, the importance of the performance evaluation tool for NICU nurses by Park and Lee [30] was 3.70 (0.30) in a previous study [2] that targeted NICU nurses in Korea. Moreover, the NICU Developmental Supportive Nursing Importance Tool, used by Hong and Son [31] showed a score of 3.77 (0.74) in another study [26], obtaining results similar to those of this study, even though the performance was relatively lower in the two previous studies [2,26] at 3.50 (0.35) and 3.46 (0.81), respectively. The difference appears to have arisen in Lim et al. [2] due to measuring performance converted into performance confidence. Another previous study [19] measured the importance of newborn care, targeting Tanzania’s registered nurses. The results demonstrated 92.6–98.7 out of 100 points, which—converted—is 4.63–4.94 points out of 5 points. These results showed a higher perception of importance compared to the current study. The performance in the Tanzanian study was 50.0–85.6 out of 100 points, which—converted—is 2.50–4.28 points out of 5 points, demonstrating a lower performance than the current study. These results may arise from differences in perception and performance, depending on the country’s level of healthcare and the presence and experience of qualified continuous professional training. In the current study, it was confirmed that a high proportion of participants (90.9%) had received clinical nursing training for high-risk premature infants, while Mbekenga et al. [19] noted that there were issues such as a lack of quality education and training for healthcare providers, a lack of knowledge and skills among healthcare providers in neonatal nursing, and an unacceptably high neonatal mortality ratio.

Analysis of the differences in nursing importance and performance according to general characteristics shows that importance and performance were significantly higher in the older age group with higher final academic degree and longer clinical, nursery, and NICU experience. Previous studies [2,32,33] also confirm that the longer the clinical experience, the higher the perceived professional value of nursing, supporting the results of this study. The accumulation of clinical experience seemingly prompts a better perception of the importance of the work and its professional value. This indicates that, to raise the perception of importance, nurses should be provided with sufficient experience through training on the clinical nursing of high-risk premature infants. It has been noted that greater clinical experience and higher competence demonstrates higher performance, as nursing high-risk premature infants requires special and precise nursing [2]. Since the lack of NICU medical experts threatens the safety and quality of neonatal intensive care, neonatal intensive care capacity must be strengthened through in-depth educational support [34].

This study demonstrated that greater importance was perceived in cases where clinical nursing training for high-risk premature infants was received through lectures as opposed to through simulation, case-based learning, and problem-based learning. In cases where it was received through theoretical learning methods, such as lecture-style learning, the importance was more greatly perceived due to the feeling that clinical nursing training was not sufficiently conducted. It was noted that clinical nursing training must provide an opportunity to put theoretical knowledge into practice through empirical learning [35,36].

We found that a significantly higher importance was perceived by respondents who answered that XR-based neonatal nursing training was necessary than by those who answered that it was not necessary. Previous studies [37] also demonstrate that the higher the learning motivation, the higher the need for online and remote learning, supporting the results of this study.

The domain with highest priority in training needs for nursing high-risk premature infants was the “treatment and procedure” domain, in which the “preparing and nursing for tracheostomy insertion” item demonstrated the highest need. In a previous study [2], nursing intervention involving the need for artificial ventilator was the second highest. Such needs appear to have been perceived because the respiratory systems of premature infants are often immature and physiologically unstable [26]. This study was funded with pure research support from a national institution, and has no interest in clinical institutions; funding is not a contributing factor to this result.

In the IPA analysis, items corresponding to the “concentrate here” domain were identified as “maintenance of various tubes, etc.” (No. 13) and “blood sugar test” (No. 14), requiring the reinforcement of assessment competence. In previous studies [2], both the importance and performance of high-risk neonatal assessment were high, supporting the results of this study. The assessment of physical conditions is seemingly perceived to be important in terms of potential risk management, including assessment before, during, and after neonatal nursing.

Upon identifying overlapping items by aggregating the IPA, Borich priority formula, and the locus for focus model to ascertain priorities in the nursing training needs for high-risk premature infants, the items of topmost priority included “maintenance of various tubes” (No. 54), while items of secondary priority included “intake and output check” (No. 7), “droplet and air containment nursing” (No. 37), “blood transfusion” (No. 46), “therapeutic thermoregulation” (No. 61), and “NRP” (No. 64). Therefore, it appears that these contents should be developed as a priority in NICU simulation programs. In particular, as the importance of infectious disease prevention nursing has been emphasized since the COVID-19 pandemic, programs including infectious disease prevention nursing that block emerging infectious diseases appear to be necessary [38].

## 5. Conclusions

The opportunities for clinical training for infant patients in the nursery and NICU are gradually becoming insufficient and limited to observation-oriented training. Premature infants admitted to the NICU require specialized and highly sophisticated care. This study was conducted to identify the educational content that the nursery and NICU nurses require for additional training while nursing newborns in clinical settings. According to the study results, nursery and NICU nurses have the highest training needs for maintenance of various tubes in nursing high-risk premature infants. To strengthen the competence of nursery and NICU nurses in clinical nursing for high-risk premature infants, training programs on the maintenance of various tubes, such as the endotracheal tube, nasogastric tube, orogastric tube, foley catheter, central line catheter, chest tube, rectal tube, and peritoneal dialysis catheter, should be developed. Additionally, nursing educational training for the needs of neonatal assessment and droplet and air containment in preparation for emerging infectious diseases has also become necessary.

This study is significant in that it provided data for XR simulation programs to be developed in the future by identifying the priorities of training needs for the nursing of high-risk premature infants using various analytical frameworks. However, the nursery and NICU nurses who participated in this study were limited to a specific country, and therefore, the research results cannot be generalized. It is thus necessary to conduct additional studies in the future to understand the differences in the training needs of nurses according to regional, environmental, and economic characteristics by including participants from urban and rural areas as well as from countries of various economic levels. Furthermore, we suggest the development of training programs reflecting the research progress and verifying the effectiveness thereof.

## Figures and Tables

**Figure 1 healthcare-10-02197-f001:**
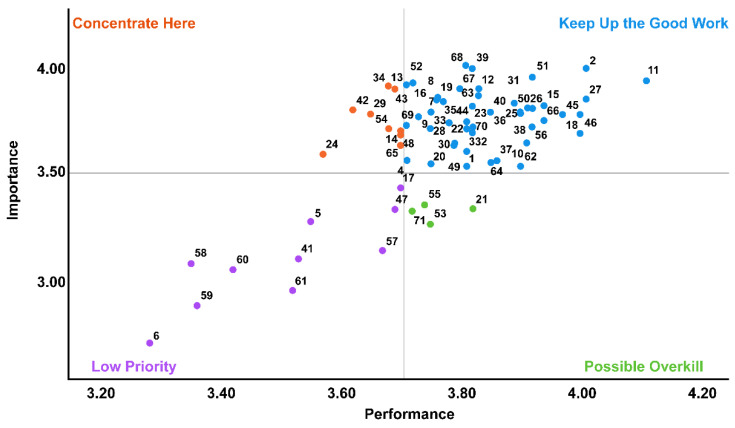
Importance-Performance Analysis (IPA) matrix of nursing of high-risk premature infants in NICU (Refer to Table 2 for items 1 to 71).

**Figure 2 healthcare-10-02197-f002:**
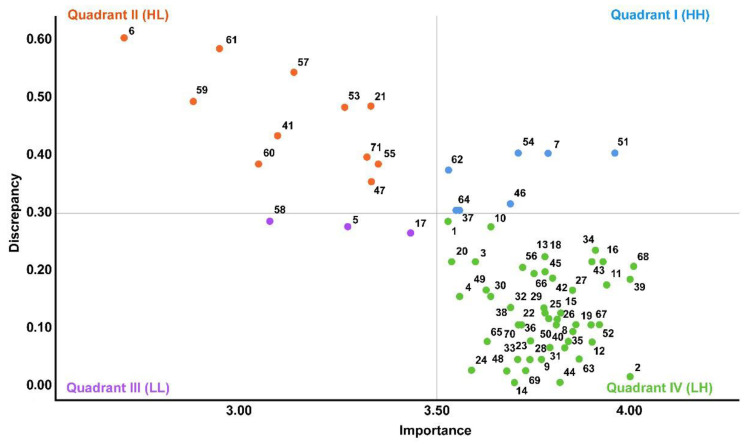
The locus for focus model of nursing of high-risk premature infants in NICU (Refer to Table 3 for items 1 to 71).

**Table 1 healthcare-10-02197-t001:** Differences in Importance of and Performance in Neonatal Intensive Care according to General Characteristics (N = 99).

Characteristics	Categories	n (%)	Importance	*t* or F	*p*-Value	Performance	*t* or F	*p*-Value
			Mean (SD)			Mean (SD)		
Age group (years)	20s	12 (12.1)	3.75 (0.48) a	9.87	<0.001	3.76 (0.53) a	10.35	<0.001
	30s	68 (68.7)	3.48 (0.45) b	c, d > b		3.59 (0.61) b	c > a, b, d	
	40s	13 (13.1)	4.08 (0.59) c	(Scheffé)		4.47 (0.51) c	(Scheffé)	
	50s	6 (6.1)	4.22 (0.30) d			4.33 (0.44) d		
Final academic degree	Bachelor	82 (82.8)	3.53 (0.47)	−4.69	<0.001	3.63 (0.60)	−5.36	<0.001
	Master	17 (17.2)	4.13 (0.52)			4.46 (0.46)		
Clinical experience (years)	More than 1~less than 3	2 (2.0)	3.43 (0.17)	14.67	<0.001	3.43 (0.33)	19.284	<0.001
	More than 3~less than 5	5 (5.1)	3.42 (0.50)	b > a		3.69 (0.52)	b > a	
	More than 5~less than 7	52 (52.5)	3.40 (0.31) a	(Scheffé)		3.42 (0.40) a	(Scheffé)	
	More than 7	40 (40.4)	4.00 (0.57) b			4.25 (0.66) b		
Clinical experience in the nursery/NICU (year)	less than 1	3 (3.0)	3.50 (0.17) a	15.61	<0.001	3.67 (0.48) a	17.74	<0.001
	More than 1~less than 3	22 (22.2)	3.54 (0.44) b	e > a, b, c, d		3.61 (0.57) b	e > a, b, c, d	
	More than 3~less than 5	33 (33.3)	3.39 (0.27) c	(Scheffé)		3.46 (0.43) c	(Scheffé)	
	More than 5~less than 7	17 (17.2)	3.48 (0.41) d			3.52 (0.53) d		
	More than 7	24 (24.3)	4.21 (0.55) e			4.52 (0.53) e		
Number of assigned patients	1~3	62 (62.6)	3.70 (0.56)	1.36	0.177	3.85 (0.68)	1.72	0.089
	More than 4	37 (37.4)	3.55 (0.44)			3.63 (0.60)		
Working unit	Nursery	45 (45.5)	3.38 (0.26)	−5.26	<0.001	3.39 (0.35)	−6.45	<0.001
	NICU	54 (54.5)	3.85 (0.59)			4.08 (0.69)		
Experience in clinical nursing training for high-risk premature infants	Yes	90 (90.9)	3.62 (0.52) b	−1.04	0.300	3.81 (0.53)	3.40	0.069
	No	9 (9.1)	3.81 (0.51)			3.70 (0.63)		
Training site for the clinical nursing of high-risk premature infants ^†^	On-site	48 (53.3)	3.69 (0.57)	1.29	0.199	3.76 (0.65)	0.97	0.336
	Off-site	42 (46.7)	3.55 (0.47)			3.63 (0.61)		
High-risk preterm infant clinical nursing Training method for the clinical nursing of high-risk premature infants ^†^	Lecture	29 (32.2)	3.99 (0.61) a	11.18	<0.001	3.85 (0.65)	18.13	0.121
	Simulation	27 (30.0)	3.57 (0.42) b	a > b, c, d		3.59 (0.51)		
	Case-based learning	24 (26.7)	3.35 (0.31) c	(Scheffé)		3.33 (0.35)		
	Problem-based learning	10 (11.1)	3.33 (0.24) d			3.31 (0.24)		
Desired training site	On-site	66 (66.7)	3.58 (0.47)	−1.41	0.164	3.71 (0.62)	−1.10	0.274
	Off-site	33 (33.3)	3.75 (0.61)			3.88 (0.73)		
Confidence in clinical nursing for high-risk premature infants	Lack	4 (4.0)	3.75 (0.09)	0.13	0.876	3.77 (0.28)	0.01	0.993
	Moderate	26 (26.3)	3.61 (0.46)			3.76 (0.66)		
	High	69 (69.7)	3.64 (0.52)			3.77 (0.68)		
Need XR training on neonatal care	Yes	96 (97.0)	4.34 (0.51)	−2.40	0.018	3.81 (0.17)	−1.31	0.121
	No	3 (3.0)	3.62 (0.52)			3.74 (0.65)		

Notes. ^†^ 90 subjects with clinical nursing education experience for high-risk premature infants; NICU = Neonatal intensive care unit; XR = Extended Reality (including virtual reality, augmented reality, and mixed reality technologies); a. b, c, d, e, Different lowercase letters in lines indicate the statistical analysis results of the mean differences, and scores are summarized.

**Table 2 healthcare-10-02197-t002:** Importance-Performance Analysis (IPA) and Borich Analysis of Nursing of High-risk Prematurity in NICU (N = 99).

No.	Contents	Importance	Performance	Gap	Paired *t*-Test	*p*-Value	Borich Needs	Rank
Total		3.78 (0.67)	3.69 (0.57)	0.09 (0.29)	3.31	0.001	3.63	
Monitoring and measuring		3.71 (0.68)	3.56 (0.50)	0.15 (0.43)	3.45	0.001	0.56	2
1	Measurement of body temperature, respiration, and heart rate	3.81 (1.27)	3.53 (1.37)	0.28 (1.20)	2.35	0.021	1.07	19
2	Device monitoring	4.01 (0.97)	4.00 (0.98)	0.01 (1.04)	0.10	0.928	0.04	69
3	Invasive pressure monitoring	3.81 (1.35)	3.60 (1.38)	0.21 (1.79)	1.18	0.240	0.80	30
4	Monitoring others	3.71 (1.11)	3.56 (1.10)	0.15 (1.40)	1.08	0.283	0.56	41
5	Installation of a monitor for pressure measurement	3.55 (1.17)	3.27 (1.12)	0.27 (1.34)	2.03	0.045	0.96	22
6	Maintenance of IABP or ECMO	3.28 (1.44)	2.69 (1.38)	0.60 (1.76)	3.37	0.001	1.97	3
7	Intake & output check	3.79 (1.32)	3.75 (1.34)	0.40 (1.41)	0.29	0.776	1.52	9
8	Weighing high-risk premature infants	3.85 (1.30)	3.76 (1.28)	0.09 (1.46)	0.62	0.536	0.35	55
9	Body measurements	3.77 (1.17)	3.73 (1.19)	0.04 (1.14)	0.35	0.726	0.15	64
Physical examination and lab test		3.82 (0.66)	3.78 (0.62)	0.04 (0.38)	1.03	0.306	0.15	8
10	State of consciousness observation	3.91 (1.15)	3.64 (1.24)	0.27 (1.41)	1.92	0.058	1.06	20
11	Auscultation and recording of breath and heart sounds	4.11 (1.04)	3.94 (1.06)	0.17 (1.20)	1.42	0.159	0.70	36
12	Circulation/sensation/motion check	3.90 (1.06)	3.83 (1.16)	0.07 (1.33)	0.53	0.597	0.27	56
13	Other circumstances (fall, pain, sores, sedation, etc.)	3.91 (1.15)	3.68 (1.26)	0.23 (1.38)	1.68	0.096	0.90	24
14	Blood sugar test	3.70 (1.27)	3.70 (1.29)	0.00 (1.45)	0.00	1.000	0.00	70
15	Emergency lab test (Stat)	3.94 (1.10)	3.82 (1.17)	0.12 (1.15)	1.05	0.299	0.47	44
16	Collecting other samples (sputum, urine, feces, etc.)	3.93 (1.10)	3.72 (1.13)	0.21 (1.18)	1.79	0.077	0.83	27
17	3-lead ECG recording	3.70 (1.15)	3.43 (1.16)	0.26 (1.42)	1.84	0.068	0.96	21
Respiratory care		3.81 (0.78)	3.70 (0.67)	0.11 (0.42)	2.55	0.012	0.42	5
18	Maintain O_2_ therapy	4.00 (1.18)	3.78 (1.18)	0.22 (1.27)	1.74	0.086	0.88	25
19	Initiation and exchange O_2_ therapy	3.86 (1.15)	3.76 (1.27)	0.10 (1.40)	0.72	0.473	0.39	51
20	Maintain NO therapy	3.75 (1.26)	3.54 (1.21)	0.21 (1.72)	1.23	0.223	0.79	31
21	Initiation and exchange NO therapy	3.82 (1.23)	3.33 (1.22)	0.48 (1.42)	3.39	0.001	1.83	4
22	Apnea care	3.81 (1.29)	3.71 (1.26)	0.10 (1.33)	0.76	0.451	0.38	53
23	Chest physiotherapy	3.81 (1.16)	3.74 (1.16)	0.07 (1.40)	0.50	0.617	0.27	57
24	Apply nebulizer	3.57 (1.20)	3.59 (1.18)	0.02 (1.45)	0.14	0.890	0.07	68
25	Airway aspiration	3.90 (1.20)	3.78 (1.18)	0.12 (1.23)	0.98	0.330	0.47	45
26	Maintaining ventilator (invasive and non-invasive)	3.91 (1.17)	3.81 (1.18)	0.10 (1.40)	0.72	0.476	0.39	49
27	Ventilator setting and application	4.01 (1.18)	3.85 (1.13)	0.16 (1.34)	1.20	0.232	0.64	38
28	Intubation	3.78 (1.33)	3.74 (1.16)	0.04 (1.53)	0.26	0.793	0.15	63
29	Extubation	3.78 (1.16)	3.65 (1.21)	0.13 (1.47)	0.89	0.376	0.49	43
30	Artificial airway (endotracheal tube) care	3.79 (1.26)	3.64 (1.13)	0.15 (1.33)	1.14	0.259	0.57	40
Mobility		3.85 (1.00)	3.76 (1.07)	0.10 (1.13)	0.84	0.402	0.39	6
31	Position change	3.89 (1.18)	3.83 (1.32)	0.06 (1.35)	0.45	0.657	0.23	60
32	Rehabilitation	3.82 (1.17)	3.69 (1.20)	0.13 (1.50)	0.87	0.385	0.50	42
Hygiene and infection control		3.79 (0.78)	3.76 (0.79)	0.03 (0.11)	0.60	0.547	0.11	10
33	Bathing	3.75 (1.24)	3.71 (1.26)	0.04 (1.40)	0.29	0.774	0.15	65
34	Exchange linen	3.68 (1.18)	3.91 (1.16)	0.23 (1.34)	1.73	0.087	0.85	26
35	Incubator management	3.77 (1.21)	3.84 (1.27)	0.07 (1.37)	0.51	0.609	0.26	58
36	Contact quarantine and counter-isolation nursing	3.90 (1.17)	3.79 (1.22)	0.11 (1.26)	0.88	0.383	0.43	48
37	Droplet and air containment nursing	3.86 (1.16)	3.56 (1.31)	0.30 (1.52)	1.98	0.050	1.16	17
Feeding		3.89 (1.11)	3.86 (1.04)	0.03 (0.50)	0.31	0.758	0.12	9
38	Gavage-tube feeding	3.90 (1.22)	3.78 (1.36)	0.12 (1.28)	0.94	0.348	0.47	46
39	Bottle feeding	4.00 (1.17)	3.82 (1.18)	0.18 (1.27)	1.43	0.156	0.72	35
Elimination		3.66 (0.82)	3.56 (0.70)	0.10 (0.85)	1.22	0.224	0.37	7
40	Diaper change	3.85 (1.16)	3.79 (1.22)	0.06 (1.25)	0.48	0.631	0.23	61
41	Nelaton catheterization	3.53 (1.24)	3.09 (1.22)	0.43 (1.55)	2.78	0.006	1.52	8
42	Enema	3.80 (1.20)	3.62 (1.21)	0.18 (1.55)	1.17	0.245	0.68	37
Medication and transfusion		3.81 (0.79)	3.69 (0.63)	0.12 (0.54)	2.18	0.032	0.46	4
43	Insert IV route or arterial line	3.90 (1.01)	3.69 (1.28)	0.21 (1.25)	1.69	0.094	0.82	28
44	Continuous fluid infusion via infusion/syringe pump	3.82 (1.26)	3.82 (1.18)	0.00 (1.26)	0.00	1.000	0.00	71
45	Intravenous injection	3.97 (1.18)	3.78 (1.23)	0.19 (1.38)	1.38	0.170	0.75	33
46	Blood transfusion	4.00 (1.20)	3.69 (1.20)	0.31 (1.54)	2.02	0.046	1.24	16
47	Exchange transfusion	3.69 (1.14)	3.33 (1.27)	0.35 (1.68)	2.09	0.039	1.29	15
48	surfactant administration (Intratracheal)	3.70 (1.24)	3.68 (1.19)	0.02 (1.57)	0.13	0.898	0.07	67
49	Non-intravenous medication	3.79 (1.26)	3.63 (1.24)	0.16 (1.39)	1.16	0.250	0.61	39
Treatment and procedure		3.70 (0.66)	3.43 (0.54)	0.27 (0.67)	4.06	<0.001	1.00	1
50	Tube insertion	3.92 (1.22)	3.81 (1.21)	0.11 (1.50)	0.74	0.462	0.43	47
51	Simple dressing	3.96 (1.07)	3.92 (1.08)	0.40 (1.25)	0.32	0.747	1.58	7
52	Tube irrigation and instillation	3.71 (1.26)	3.92 (1.08)	0.10 (1.23)	1.71	0.090	0.37	54
53	Prepare puncture	3.75 (1.15)	3.26 (1.16)	0.48 (1.50)	3.21	0.002	1.80	5
54	Maintenance of various tubes	3.71 (1.18)	3.68 (1.19)	0.40 (1.41)	0.29	0.776	1.48	10
55	Complex dressing	3.74 (1.22)	3.35 (1.29)	0.38 (1.56)	2.44	0.016	1.42	13
56	Phototherapy	3.92 (1.12)	3.72 (1.25)	0.20 (1.28)	1.57	0.119	0.78	32
57	Preparing and nursing for tracheostomy insertion	3.67 (1.26)	3.13 (1.36)	0.54 (1.88)	2.84	0.005	1.98	2
58	Observe intracranial pressure monitoring device	3.35 (1.22)	3.07 (1.27)	0.28 (1.69)	1.67	0.099	0.94	23
59	Preparing for CRRT insertion	3.36 (1.32)	2.87 (1.37)	0.49 (1.80)	2.73	0.007	1.65	6
60	CRRT maintenance care	3.42 (1.28)	3.04 (1.38)	0.38 (1.68)	2.28	0.025	1.30	14
61	Perform peritoneal dialysis	3.52 (1.35)	2.94 (1.31)	0.58 (1.77)	3.24	0.002	2.04	1
62	Therapeutic thermoregulation	3.90 (1.11)	3.53 (1.22)	0.37 (1.45)	2.56	0.012	1.44	12
63	Other treatment and exam (X-ray, ultrasound, etc.)	3.87 (1.22)	3.83 (1.15)	0.04 (1.29)	0.31	0.755	0.15	62
64	NRP	3.85 (1.26)	3.55 (1.25)	0.30 (1.69)	1.78	0.078	1.16	18
Emotional support, communication, and education		3.82 (1.05)	3.69 (1.02)	0.13 (1.02)	1.28	0.205	0.50	3
65	Family Counseling	3.70 (1.22)	3.63 (1.34)	0.07 (1.44)	0.39	0.697	0.26	59
66	Developmental supportive care	3.94 (1.19)	3.75 (1.03)	0.19 (1.31)	1.21	0.228	0.75	34
Admission and discharge management		3.77 (0.87)	3.74 (0.71)	0.03 (0.63)	0.54	0.588	0.11	11
67	Admission	3.90 (1.24)	3.80 (1.24)	0.10 (1.39)	0.72	0.471	0.39	50
68	Discharge	4.01 (0.92)	3.81 (1.29)	0.20 (1.35)	1.49	0.139	0.80	29
69	Transfer preparation	3.73 (1.31)	3.71 (1.26)	0.02 (1.46)	0.14	0.891	0.07	66
70	Discharge education (for caregiver)	3.82 (1.21)	3.72 (1.24)	0.10 (1.32)	0.76	0.449	0.38	52
71	Care of the dying patient	3.72 (1.23)	3.32 (1.30)	0.39 (1.68)	2.34	0.021	1.45	11

Notes. IABP, intra-aortic balloon pump; ECMO, extracorporeal membrane oxygenator; O_2_, oxygen; NO, nitrogen; CRRT, continuous renal replacement therapy; NRP, neonatal resuscitation program.

**Table 3 healthcare-10-02197-t003:** Combining Borich Needs and the Locus for Focus Model to Derive Priorities for High-risk Premature Nursing Training (N = 99).

Domain	Item	Contents	IPA	Borich Needs	The Locus for Focus	Priority
MM	6	Maintenance of IABP or extracorporeal membrane oxygenator (ECMO)	LP	3	HL	3
	7	Intake & output check	KU	9	HH	2
RC	21	Initiation and exchange NO therapy	PO	4	HL	3
HIC	37	Droplet and air containment nursing	KU	17	HH	2
EM	41	Nelaton catheterization	LP	8	HL	3
MTTP	46	Blood transfusion	KU	16	HH	2
	47	Exchange transfusion	LP	15	HL	3
TP	51	Simple dressing	KU	7	HH	2
	53	Prepare puncture	PO	5	HL	3
	54	Maintenance of various tubes	CH	10	HH	1
	55	Complex dressing	PO	13	HL	3
	57	Preparing and nursing for tracheostomy insertion	LP	2	HL	3
	59	Preparing for CRRT insertion	LP	6	HL	3
	60	CRRT maintenance care	LP	14	HL	3
	61	Perform peritoneal dialysis	LP	1	HL	3
	62	Therapeutic thermoregulation	KU	12	HH	2
	64	NRP	KU	18	HH	2
ADM	71	Care of the dying patient	PO	11	HL	3

Notes. Domain: MM = monitoring and measuring; RC = respiratory care; HIC = hygiene and infection control; EM = elimination; MTTP = medication and transfusion; TP = treatment and procedure; ADM = admission and discharge management. IPA = importance-performance analysis: CH = concentrate here; KU = keep up the good work; LP = low priority; PO = possible overkill. The locus for focus: HL = high discrepancy, low importance; HH = high discrepancy, high importance; LL = low discrepancy, low importance; LH = low discrepancy, high importance.

## Data Availability

Not applicable.

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
