# Peer review of "Simulation Training Needs of Nurses for Nursing High-Risk Premature Infants: A Cross-Sectional Study"

_healthcare, 2022, doi:10.3390/healthcare10112197_

Round 1

Reviewer 1 Report

This paper highlighted the insufficiency of the current NICU nursery training. In fact, actually the clinical training for infant patients is basically an observation-oriented training. 

From a brief review of the current literature, the topic of medicine education science is poorly explored. So, I think this article is denoted by a good originality.

The text is very specialistic. So, it is not fully clear to read, but it is appropriate for a scientific publication.

The conclusions elaborated a good data processing that led to an appropriate conclusion that is well presented. In fact, some procedures are not learned by nurses after school trainingship, so these procedures must be introduced in the e-learning modules. 

Author's conclusion are appropriated. NICU nurses training is important and a XR-based neonatal nursing training programs should be added in the degree and post-degree educational program.   Minor revisions are needed.

Please provide fac-simile of the survey and the order of questions, in order to make reader aware of possible bias,

well done, 

have a nice work  

Author Response

Please see the responses attached

Reviewer 2 Report

The authors aim to determine the differences and priorities in the training needs and performance of newborn nursery and NICU nurses for nursing high-risk premature infants. Overall, the authors have gathered sufficient data, and their scientific conclusions align with the data presented. A few suggestions to improve the manuscript are as follows.
1) Is there any gender bias in these observations? Both male and female nurses were included in this study? Please add details in the methods sections. 

2) “maintenance of various tubes showed the highest priority." This conclusion was apt based on the data collected, but authors might add hospital funding as a variable or at least mention in the discussion that this was not/was a supposedly contributing factor. 

Author Response

Please see the response attached
